# Non-Targeted Metabolomics Reveals Sorghum Rhizosphere-Associated Exudates are Influenced by the Belowground Interaction of Substrate and Sorghum Genotype

**DOI:** 10.3390/ijms20020431

**Published:** 2019-01-19

**Authors:** Sarah B. Miller, Adam L. Heuberger, Corey D. Broeckling, Courtney E. Jahn

**Affiliations:** 1Bioagricultural Sciences and Pest Management, Colorado State University, Fort Collins, CO 80523, USA; sbm3@g.clemson.edu; 2Horticulture and Landscape Architecture, Colorado State University, Colorado State University, Fort Collins, CO 80523, USA; Adam.Heuberger@colostate.edu; 3Proteomics and Metabolomics Facility, Colorado State University, Colorado State University, Fort Collins, CO 80523, USA; Corey.Broeckling@ColoState.EDU

**Keywords:** GC-MS, LC-MS, metabolomics, root exudate, rhizosphere, sorghum

## Abstract

Root exudation is an important plant process by which roots release small molecules into the rhizosphere that serve in overall plant functioning. Yet, there is a major gap in our knowledge in translating plant root exudation in artificial systems (i.e., hydroponics, sterile media) to crops, specifically for soils expected in field conditions. Sorghum (*Sorghum bicolor* L. Moench) root exudation was determined using both ultra-performance liquid chromatography and gas chromatography mass spectrometry-based non-targeted metabolomics to evaluate variation in exudate composition of two sorghum genotypes among three substrates (sand, clay, and soil). Above and belowground plant traits were measured to determine the interaction between sorghum genotype and belowground substrate. Plant growth and quantitative exudate composition were found to vary largely by substrate. Two types of changes to rhizosphere metabolites were observed: rhizosphere-enhanced metabolites (REMs) and rhizosphere-abated metabolites (RAMs). More REMs and RAMs were detected in sand and clay substrates compared to the soil substrate. This study demonstrates that belowground substrate influences the root exudate profile in sorghum, and that two sorghum genotypes exuded metabolites at different magnitudes. However, metabolite identification remains a major bottleneck in non-targeted metabolite profiling of the rhizosphere.

## 1. Introduction

The phenotypic plasticity of plant root systems allows for modification in their morphology, physiology, and/or biochemistry to physical, chemical, and biological changes in the belowground environment [1,2]. Root exudates, chemical compounds released from the roots into the adjacent soil (the rhizosphere), are a critical component of this response [3]. These versatile exudates serve many purposes, including facilitating water and nutrient acquisition, mediating positive and negative microbial symbioses, and functioning as natural pesticides and herbicides [4,5]. The composition of these root exudates is highly variable, varying both quantitatively and qualitatively to changes in the environment as well as varying among plant species, genotypes, and even plant developmental stages [3,6,7]. Thus, the potential to utilize exudate variation is a promising tool in both plant breeding and agronomic practices as it represents an opportunity to reduce the application of costly chemical inputs such as fertilizers, herbicides, and pesticides [5,8,9,10].

Root exudation can either directly or indirectly improve plant fitness to help mitigate stressful conditions [7]. For instance, when soils become compacted or dry, roots can secrete viscous mucilage to promote root growth, which increases the plant’s ability to acquire water and nutrients [1]. Plants can also improve fitness indirectly by exuding metabolites that recruit specific plant growth-promoting microorganisms (hereafter PGPM) in the rhizosphere that are beneficial in the given environment [11,12]. These PGPM can help buffer against extreme conditions by acquiring trace nutrients, regulating hormone production, or by acting as biological controls to defend against pathogens [13]. Exploring how the plant interacts with its physical, chemical, and biological environment can therefore help to understand the specific roles of root exudates. Furthermore, this knowledge can potentially be implemented in sustainable agricultural practices through plant variety selection, crop rotation, or biochemical soil inoculations.

Many studies have observed root exudation by evaluating plants or individual plant-microbe interactions in artificial conditions (e.g., hydroponic systems or sterile media), providing a baseline of knowledge for this belowground occurrence [14]. However, belowground interactions between the plant and its abiotic and biotic environment are much more dynamic in agricultural settings [9]. For example, the amount of root exudation is influenced by the microbial presence and inherent soil properties [7].

Several physical and chemical characteristics of the soil such as structure, pH, and previous plant cultivation greatly influence the amount of nutrient availability and impact root growth, exudation, and the microbial presence [7,15,16,17]. Sterilizing the substrate by autoclaving alters these inherent soil properties including macronutrient availability, soil aggregation, and organic matter structure, thus influencing patterns of root exudation [18,19]. For instance, sterilizing soils can increase nutrient adsorption, increasing the exudation of chemical compounds such as chelators to bind nutrients [20]. Additionally, root exudation increases in the presence of microorganisms due to microbial consumption and turnover, while sterile systems using artificial media or autoclaved soils likely underestimate the rate of root exudation in comparison to natural systems [7,21]. Therefore, by utilizing and characterizing substrates that represent soil ecosystems, we can determine the ecological significance of root exudation to improve crop production.

Root exudation differs in qualitative composition and quantity of several different classes of metabolites (small molecules formed from plant metabolism) including carbohydrates, amino acids, organic acids, vitamins, secondary metabolites, and high molecular weight compounds such as mucilage [3,22]. It is estimated that 200 plant-biosynthesized compounds can be released as root exudates [23]. However, many root exudate studies target single metabolites or groups of metabolites, such as the case with the root exudate sorgoleone produced by the crop species, sorghum (*Sorghum bicolor* L. Moench). Sorgoleone is an allelopathic root exudate that has been studied in sorghum for its genotypic variation and its mechanism for weed suppression [24,25,26]. Yet, sorghum is a crop species that is noted for its adaption to drought and heat and it is unknown if root exudation of this species contributes towards these tolerances. Therefore, future studies should evaluate the broad spectrum of exudates that are produced in response to environmental conditions that may aid in the plant’s success.

Most root exudates are low molecular weight compounds that are products of both primary or specialized plant metabolism [27]. Therefore, metabolomics is an attractive method to characterize how genetic and environmental factors influence root exudation. Plant metabolomics is often performed using gas chromatography-mass spectrometry (GC-MS) and/or ultra-performance liquid chromatography-mass spectrometry (UPLC-MS) [6,28,29], with each of these platforms having their own strengths and limitations [30]. Although the progression of the metabolomics field to identify and quantify compounds is rapidly occurring with an increasing number of standards and improving methodologies [31,32], metabolite annotation remains a major bottleneck in non-targeted metabolomics [33]. Nevertheless, the use of non-targeted metabolomics in plant biology to understand genotypic effects on metabolite variation is becoming more common, ranging from applications in stress physiology to food quality [34]. The use of non-targeted metabolomics across multiple platforms will identify a broad range of metabolites in the rhizosphere to determine the root exudate profile.

In this study, we assessed metabolites enriched by the plant’s rhizosphere (rhizosphere-associated metabolites). Our overall goal was to determine if plant growth and rhizosphere-associated metabolites varied between sorghum genotypes and among substrates that differed in physico-chemical properties. We utilized non-targeted metabolomics and both GC- and UPLC-MS platforms to ascertain the ability of each platform to extract metabolites from the rhizosphere. Furthermore, we evaluate the viable microbial presence in the rhizosphere of each genotype in each substrate to further assess the exudate profile. Taken together, our results indicate a robust method to evaluate genotypic exudate variation in response to various environmental conditions.

## 2. Results

### 2.1. Soil Characteristics and Viable Microbial Presences Vary Among Substrates

Three substrates (clay, sand, and soil) differing in physico-chemical properties were utilized to compare plant growth and rhizosphere-associated metabolites in sorghum (see Appendix A for soil properties). Two sorghum genotypes were evaluated within each substrate. To assess metabolites enriched by the plant’s rhizosphere, controls within each substrate did not contain a plant (no-plant controls) and were designed to distinguish metabolites that were characteristic of the bulk substrate, and therefore determine which metabolites were rhizosphere-associated. We termed exudates as rhizosphere-associated as they may encompass both plant and microbial exudates. Substrates were not autoclaved as the heat, steam, and pressure are expected to alter substrate characteristics [18,19,20].

We additionally determined the microbial presence for each treatment and substrate. When comparing the no-plant controls of the three substrates, the highest number of viable bacteria was detected in the soil, followed by clay and then sand (Figure 1). Within soil, the SC56 plant treatment had a slightly lower microbial presence than the no-plant control. Within the clay and sand substrates, both plant treatments had substantially greater viable microbial counts than respective no-plant controls. Among substrates, both genotypes kept a relatively consistent microbial presence. However, the microbial presence for the SC56 plant treatment displayed lower levels than that of BTx623 within each substrate.

### 2.2. Variation in Plant Morphology is Largely Influenced by Substrate

To understand how substrates influence sorghum’s allocation of resources to above and below ground traits, sorghum plants were grown in three substrates for 21 days, after which leaf areas and several root traits were measured (Figure 2).

Leaf areas were smaller for plants grown in sand and clay than plants grown in soil (*p* < 0.0001), and there were no differences between sorghum genotypes (Figure 2a). Substrate also affected root morphology (Figure 2b,c). Plants grown in sand had the shortest total root lengths (*p* < 0.0001) and largest average root diameters (*p* < 0.0001), and this effect was comparable across genotypes. Total root lengths and average root diameters were more similar between plants grown in clay and soil in comparison to those grown in sand. However, genotype BTx623 had longer total root lengths than SC56 in soil, while genotype SC56 had larger average root diameters than those of BTx623 in both clay and soil substrates. Overall, plants grown in sand had smaller above and below ground biomass investments than plants grown in clay or soil.

### 2.3. Non-Targeted Metabolomics Detected Rhizosphere-Enhanced or -Abated Metabolites

We detected metabolites using a non-targeted metabolomics approach. The GC- and UPLC-MS analyses resulted in 34,718 and 2929 molecular features that were deconvoluted into an estimated 829 and 475 compounds, respectively. The metabolomics data was evaluated to compare trends in the root-exuded metabolite profiles using principal component analysis (PCA) on the total 1304 compounds. Four principle components (PCs) explained 64% of the variation. Principle Component 1 (28.1%) and PC3 (10.6%) explained variation associated with substrate and plant treatment (i.e., the effect of the plant present in the substrate) (Figure 3a), respectively. The PCs separated by substrate (PC1, soil and clay/sand) and plant treatment (PC3, BTx623/SC56 and Control). Principle Component 4 also displayed variation attributed to substrate (7.5%) (clay and soil/sand) (Figure 3b). Principle Component 2 (17.8%) was variation not attributed to plant treatment or substrate, for example potentially due to variation by plant replicates (Figure 3b). The PCA supports that overall variation in metabolites (i.e., the type of metabolites, and the abundance of the metabolite) is influenced by both substrate and plant treatment.

Individual metabolites that varied due to each plant genotype (BTx623 and SC56) and substrate were determined by an ANOVA conducted within each substrate (FDR adjusted *p* < 0.05) (data not shown). Additionally, each plant treatment (BTx623 and SC56) was evaluated for metabolites that increased or decreased compared to the no-plant control within each substrate. Metabolites that changed by ±2-fold (plant treatment/no-plant control) were considered changing within the system. Changes that were 2-fold or greater were considered rhizosphere-enhanced metabolites (REMs). Additionally, metabolites of −2-fold or less were considered diminished and are termed rhizosphere-abated metabolites (RAMs). The ANOVA *p*-values and log_2_ fold changes (FCs) between each plant treatment (BTx623 and SC56) and no-plant control for all detected metabolites are displayed as volcano plots (Appendix A). Hereafter, we will describe metabolites of interest using the term log_2_ FC to indicate the relative amounts detected between plant treatments and no-plant controls and compare across substrates.

Using *p*-values (FDR adjusted *p* < 0.05) from ANOVAs conducted within each substrate and fold change criteria (log_2_ FC > 1.0) for both sorghum genotypes, a total of 219 compounds varied across all the treatments. It was found that 73 REMs varied in clay (5.6% of the detected compounds), 105 varied in sand (8.1%), and 11 REMs varied in soil (0.8%) (Table 1). Of the REMs, only eight were common to all three substrates (Figure 4a). Clay and sand had the most shared compounds (49 compounds) and sand had the most substrate specific compounds (47 compounds). For rhizosphere-abated metabolites, 62 RAMs varied in clay (4.8%), 57 RAMs varied in sand (4.4%), and two RAMs varied in soil (0.2%) (Table 1). Sand and clay shared the highest number of RAMs with 25 compounds (Figure 4b). Clay had the largest number of substrate specific RAMs (37 compounds).

### 2.4. Annotated Metabolites Represent Known Root Exudates

A total of 42 metabolites were annotated based on matching retention time and mass spectra to in-house, external, and theoretical metabolite databases including 28 metabolites from the GC-MS and 14 metabolites from the UPLC-MS dataset (Table 2). These metabolites include carbohydrates (18), amino acids (15), organic acids (5), vitamins (1), and other metabolites (3) that are known to be root exudates.

Within each of the clay, sand, and soil substrates, sucrose was detected at the lowest levels in no-plant controls compared to plant treatments (Figure 5a). In both BTx623 and SC56 plant treatments, sucrose was detected at significantly higher levels in clay and trended to higher levels for both plant treatments in sand compared to respective no-plant controls (Figure 5a; Table 2). In clay, sucrose was found to have the highest log_2_ FCs for each plant treatment compared to those in other substrates (Table 2). Additionally, sucrose had the highest log_2_ FC compared to all other metabolites detected within the clay substrate.

Tryptophan was detected at low levels in each of the substrate’s no-plant controls (Figure 5b). In both clay and sand, tryptophan was detected in both plant treatments at significantly higher levels than their respective no-plant controls. Tryptophan was detected at the highest level in the plant treatments of the sand substrate, followed by the clay and soil substrates. The organic acid quinic acid was detected at significantly higher levels in each of the plant treatments within all of the substrates (Figure 5c). Malic acid in both plant treatments was detected at higher levels in clay (Figure 5d). However, although not significant, malic acid was detected with the highest log_2_ FC in sand (Table 2; Figure 5d).

Across no-plant controls, trehalose varied in abundance, with its lowest detected presence in the sand no-plant control (Figure 5e). Trehalose was detected with the largest log_2_ FCs in sand and was significantly different in the SC56 plant treatment although the log_2_FC also trended higher in BTx623 plant treatment within this substrate. One annotated metabolite, glycerol, was detected at significantly higher levels in the no-plant controls than both plant treatments grown in sand or clay (Figure 5f).

## 3. Discussion

This study utilized non-targeted metabolomics to investigate how differing substrate conditions and genotypic background drive variation in a broad spectrum of rhizosphere-associated metabolites in sorghum. Traditionally, root exudation is quantified by targeting select metabolites in artificial media and sterile conditions. Our approach, however, provides insight into how interactions between the genotype and both the biotic and abiotic environment, influence variation in rhizosphere-associated metabolites. This platform is especially powerful moving forward, as we can now effectively study how manipulating belowground environment (e.g., nutrient deficiencies, toxicities, microbial inoculations, exogenous biochemical applications) mediates plant–environment interactions via metabolite exudation across a variety of genotypes.

Although the effect of plant genotype on root exudation is a known occurrence largely evaluated via targeting select metabolites in artificial systems [35,36,37], our study is one of the first to determine genotypic variation in a broad range of metabolites in more realistic substrates. Furthermore, using various growth substrates and non-targeted metabolomics with both GC- and UPLC-MS, we found quantitative differences in metabolites among not just genotypes, but also substrates. Variation in root exudation in response to growth substrates has been previously observed; a single variety of lettuce (*Lactuca sativa*) grown in three substrates differing in previous plant cultivation exhibited quantitative differences in root-exuded metabolites between the substrates [16]. Similar to our study, Neumann et al. [16] annotated 33 metabolites across the substrates using the GC-MS platform, representing various amino acids, sugars, and organic acids that are known to be root exudates. In our study, we annotated metabolites that are known root exudates, and we additionally quantified their presence by comparing the plant treatments to no-plant controls. We also determined metabolites that were not only enhanced in the rhizosphere, but also quantified metabolites that were abated in the rhizosphere, offering a unique perspective into plant-rhizosphere dynamics.

Past studies using artificial environments (e.g., hydroponic systems, sterile media) have played an important role in identifying the function of specific root-exuded metabolites. However, using realistic substrates is critical if we wish to better understand how plants interact with their surroundings and overcome challenges within their natural habitats. Here, we illustrate our method’s utility by discussing a subset of annotated exudates in each substrate and how these metabolites may serve in their respective environments. Further work is required to confirm the functional roles of these metabolites, but our results display variation in many metabolites detected in earlier root exudate studies.

### 3.1. Rhizosphere-Associated Exudation Responds to Stressful Abiotic Conditions

Root exudates are known to fluctuate in response to environmental conditions [4]. Among the substrates, sand represented the poorest conditions for plant growth (Appendix A) and had the most detected rhizosphere-associated metabolites (Figure 4). Thus, many of the rhizosphere-associated metabolites in sand likely buffered against harsh abiotic conditions.

Mechanical impedance of the roots was highest in sand due to its high bulk density (Appendix A). While plants are known to facilitate growth within a dense substrate by limiting root growth and enlarging root diameters [38], they also increase root exudation of viscous compounds such as mucilage to reduce friction [1,38]. We found that roots had the shortest lengths and largest diameters when grown in the dense sand substrate (Figure 2). Although we were unable to annotate many of the rhizosphere-associated metabolites present in the sand environment, some are likely to help overcome mechanical impedance. Furthermore, we detected more rhizosphere-associated metabolites in the clay and sand substrates than in the soil substrate (Figure 3). We also found increased microbial presences in clay and sand substrates for plant treatments relative to their no-plant controls (Figure 1). This increased exudation of mechanically impeded roots increases the microbial presence within the rhizosphere, also aiding in nutrient acquisition [39]. Thus, an increase in the number of rhizosphere-associated metabolites in these substrates enriches microbial abundance, which should have important consequences for buffering against poor abiotic conditions.

Further, several metabolites involved in plant stress tolerance displayed higher log_2_ fold changes in the plant treatments of sand compared to other substrates. For instance, trehalose is a disaccharide common to both plants and microorganisms that is associated with abiotic stress such as drought, high salinity or extreme temperatures [13,40]. We found trehalose to be particularly enriched in the plant treatments of the sand substrate (Table 2). Additionally, organic acids are associated with buffering environmental conditions such as nutrient toxicities or deficiencies, especially in environments with a high pH such as sand (Appendix A) [4,41]. Organic acids released by the plant can also attract specific microorganisms, which in turn release organic acids in unfavorable environmental conditions to act as chelators to increase nutrient availability [42]. Quinic acid, a major organic acid in our system (Table 2), was detected with the highest log_2_ FC for each plant treatment in the sand substrate. In addition to buffering against abiotic stress, quinic acid is a precursor of many secondary metabolites [43,44], which serve several functions including growth and defense [45].

Malic acid was also detected with the highest log_2_ FC for each plant treatment in the sand substrate (Table 2; Figure 5d). This increase was not significant, likely due to the large variation between plant replicates, but, like other organic acids [42], malic acid is a known root exudate that has been implicated in attracting beneficial bacteria and improving nutrient availability [46,47]. Overall, it is likely that a portion of the un-annotated metabolites in the sand substrate includes organic acids among other metabolites that are known to directly or indirectly through microbial recruitment improve nutrient availabilities.

### 3.2. Root Exudates Serve to Enlist Plant Growth-Promoting Bacteria

We found that both plant genotypes kept a relatively consistent microbial presence across substrates despite differences across substrates in the viable microbial presences of the no-plant controls (Figure 1). A subset of microorganisms from the surrounding environment is generally enriched in the rhizosphere due to the rhizosphere effect [48]. This is likely reflected in the reduced viable microbial presences of the plant treatments in the soil substrate compared to the no-plant control of the soil substrate that contained a greater viable microbial presence. In contrast, plants in the sand and clay substrates experienced an increase in the viable microbial presence when compared to the low initial microbial presence of respective no-plant controls, suggesting a stimulation of the general microbial population from the surrounding environment in these substrates.

Sugars provide microorganisms with readily available sources of energy [49]. The increase in sucrose, glucose and fructose in plant treatments when compared to no-plant controls in both the clay and sand substrates (Table 2) may drive the observed increase in microorganisms in these substrates (Figure 1). In *Arabidopsis thaliana*, for example, exudation of sugars early in development helps enlist a general community of microorganisms [6]. However, amino and organic acids may attract more specific microorganisms that promote plant growth [50].

Once enlisted, plant growth-promoting microorganisms serve the plant by producing the growth-stimulating phytohormone auxin [51]. More than 80% of rhizosphere bacteria are estimated to produce IAA (indole-3-acetic acid), a dominant form of auxin that promotes plant growth [52]. The primary biosynthetic pathway to IAA is through tryptophan metabolism, which can be conducted by plants or soil microorganisms [53]. We found tryptophan to be present with the highest log_2_ FC in the sand substrate, followed by the clay and soil substrates (Table 2; Figure 5b). Additionally, plants grown in sand had the smallest leaf areas and root lengths (Figure 2). Plants grown in sand therefore may have increased tryptophan production to promote plant growth through auxin synthesis.

### 3.3. Metabolites Can Be Abated by the Rhizosphere Environment

Of particular interest is the ability of our methodology to determine rhizosphere-abated metabolites (RAMs). Log_2_ FC among these metabolites were not as large as some of the detected rhizosphere-enhanced metabolites, but several significant metabolites were detected in the clay and sand substrates that were lower in the plant treatments than in respective controls (Table 1). Glycerol was the only rhizosphere-abated metabolite in both clay and sand that was able to be annotated (Table 2; Figure 5f). Glycerol can be produced by plants or microorganisms to protect against osmotic stress [54,55], and can also provide carbon and energy to microorganisms [56]. However, glycerol in the rhizosphere negatively affects root growth in *A. thaliana* as it alters auxin distribution [57]. Although other studies have detected glycerol as a root exudate [6,16], our study provides the novel perspective of glycerol in the belowground plant-environment interaction. Glycerol may be produced in the bulk substrates of clay and sand by microorganisms. Furthermore, glycerol dissimilation may be occurring by both microorganisms and/or plants in the plant treatments. Thus, glycerol could serve as an energy source or to counteract its effects as root growth inhibitor.

We annotated another rhizosphere-abated metabolite in the soil substrate as a sugar alcohol (Table 2). Sugar alcohols such as sorbitol or mannitol are utilized as substrates by microorganisms and can enrich soil microbial functional diversity when added as a soil amendment [58]. As the soil no-plant control already has a high viable microbial presence (Figure 1), this sugar alcohol may be consumed by a diverse group of microorganisms in the rhizosphere of the soil substrate.

### 3.4. Rhizosphere-Associated Metabolite Detection and Analysis Considerations

In metabolomics, it is well known that the extraction and analytical methods implemented largely influences the detected metabolites [59]. When utilizing this method to determine rhizosphere-associated metabolites within a substrate, users should consider (1) the large plant replicate variation that may impact detecting changes in levels of metabolites of interest, (2) soil factors that affect the metabolite extraction/presence, and (3) the ability of the chosen platform to detect metabolites.

Using our criteria, we detected relatively few significant metabolites within the soil as compared to clay or sand, but several annotated metabolites were likely produced by the plant as evidenced in log_2_ FC (Table 2). For example, within the soil substrate, sucrose had one of the largest log_2_ FC, but was not considered significant for the SC56 plant treatment (Table 2). As sucrose is well-established within root exudate profiles, it is reasonable to conclude that it had a higher presence in both plant treatments than the no-plant controls within the soil substrate. It is likely that the large plant-to-plant variability (biological variability) contributes to the lack of significance (as we similarly found for malic acid in the sand substrate). Indeed, plant-to-plant variability has recently been found to represent a large portion of total variation in root metabolite profiles, with the amount of variation differing between different classes of metabolites (e.g., sugars, organic acids, amino acids, phenylpropanoids, flavonoids) [60]. Large numbers of replicates will therefore help maintain statistical power, particularly when analyzing a broad range of metabolites as with non-targeted metabolomics [60]. Additionally, plant-to-plant variability increases when using a higher concentration of methanol buffer [61], making it important to choose the appropriate extraction buffer concentration. Future metabolite analyses should also incorporate total root lengths to standardize total root exudation across plants of variable size.

Several intrinsic factors of the soil substrate presumably diminished the number of significant metabolites detected in this substrate. For instance, soil had high organic matter, cation exchange capacity (CEC), and initial viable microbial presence, all of which may contribute to binding and turnover of compounds (Appendix A; Figure 1). Furthermore, some rhizosphere-associated metabolites (i.e., phenylalanine) were detected at higher levels and with more variation in the bulk substrate controls of soil compared to the clay and sand controls (data not shown). Therefore, it is likely that several other metabolites were not considered significant within this substrate due to their high background levels but are still of biological interest. Although our analyses indicate that sand and clay substrates have more detected metabolites in common (Figure 2a and Figure 3), this may be due to the intrinsic properties of soil that mask the number of detected metabolites that were both significant and had a log_2_ FC greater than one. Implementing a combination of visual tools such as volcano plots with multivariate and univariate statistical analyses and z-score test statistics to determine metabolites of interest will additionally help to determine rhizosphere-associated metabolites. Advantages and disadvantages of several aspects of univariate analyses in non-targeted metabolomics profiling are reviewed in Vinaixa et al. [62].

Finally, using the UPLC-MS platform in addition to the GC-MS platform provided greater insight into a wide range of metabolites. The UPLC-MS platform detected aromatic amino acids (phenylalanine, tryptophan and tyrosine) (Table 2), which serve as precursors to many secondary metabolites and hormones that aid in plant abiotic or biotic stress tolerance [63,64,65]. Although GC-MS is an effective tool in detecting sugars and various amino and organic acids that are prevalent in the root exudate profile such as these aromatic amino acids, the inability to annotate these on the GC-MS platform in our study reflects the value of using multiple platforms. The UPLC-MS platform also identified dhurrin, a species-specific cyanogenic glycoside associated with sorghum [66]. Therefore, using both platforms allows for a more comprehensive understanding of the root exudate profile.

Several metabolites were unable to be annotated that were of interest between both platforms. However, the continual addition of metabolites to databases will contribute toward the progression of metabolite identifications. Furthermore, the root exudate profile likely contains secondary metabolites that are more specialized or species-specific such as allelopathic compounds juglone exuded by black walnut or sorgoleone exuded by sorghum [2]. As these metabolites are not as commonly quantified as sugars and amino and organic acids that are prevalent throughout metabolomics studies, the development of standards is required to annotate these secondary metabolites and their derivatives. As the field of metabolomics continues to advance, the identification and quantification of these metabolites can be integrated into systems biology to provide a more mechanistic understanding of plant metabolism.

## 4. Materials and Methods

### 4.1. Plant Cultivation

Two grain sorghum (*Sorghum bicolor* L. Moench) genotypes were utilized for this study due to their importance in breeding programs. BTx623 is a sequenced genotype that is pre-flowering drought tolerant [67,68], whereas SC56 is a pre-flowering drought susceptible genotype [69]. After seed germination on filter paper with fungicide solution (Maxim XL, Syngenta, Greensboro, NC, USA) contained within Petri dishes, seedlings were transplanted into 1.4-liter pots containing one of three different substrates and grown in a greenhouse experiment (30 °C day/ 23 °C night; 50% relative humidity; 12-hour photoperiod with supplemental lighting). Substrates included an all-purpose potting mix (Fafard^®^ 4P, Sun Gro Horticulture, Agawam, MA, USA), fritted clay (Field & Fairway^TM^, Profile Products LLC, Buffalo Grove, IL, USA), or sand (Quikrete^®^, The Quikrete Companies, Atlanta, GA, USA), hereafter referred to as soil, clay, and sand, respectively. Each pot was lined with muslin cloth, filled with substrate, soaked in water overnight, drained for one hour and weighed previous to seedling transplanting to determine 100% field capacity (FC). All pots were watered every other day to 100% FC and fertilized weekly by watering with 75% Hoagland’s solution to 100% FC which consisted of: KH_2_PO_4_; KNO_3_; Ca(NO_3_)_2_; MgSO_4_; H_3_BO_3_; MgCl_2_-4H_2_O; ZnSO_4_-7H_2_O; CuSO_4_-5H_2_O; MoO_3_-H_2_O; and Sequestrene 138 iron chelate.

### 4.2. Experimental Design

Five replicates for each genotype within a substrate were grown for 21 days after sowing (DAS), hereafter referred to as plant treatments. In addition, five replicates of bulk substrate containing no plant (no-plant control) for each of the substrates were maintained during that period by watering and fertilizing the same as the plant treatments and serving as no-plant controls. Plants were grown in a randomized complete block design and morphological and physiological traits were assessed in addition to root exudation.

### 4.3. Characterization of Soil Properties and Quantitative Estimation of Viable Soil Microorganisms

To determine soil properties (Appendix A), 50-gram substrate samples from the bulk substrates were mixed and sent to Ward Laboratories, Inc. (Kearney, NE, USA). To estimate the viable microbial presence, five-gram substrate samples from the rhizosphere of each replicate containing a plant or the bulk soil of the no-plant control were taken and placed into 45 mL of 0.85% sterile saline solution. Samples were mixed for one minute and the solution was allowed to settle. Serial dilutions were completed and transferred to 10% tryptic soy broth plus 1.5% agar plates. Plates were incubated at 28 °C and colony forming units (CFUs) were counted daily. Counts were then calculated by multiplying CFU by the dilution factor and soil moisture to obtain the total number of microorganisms/g of dry soil.

### 4.4. Assessment of Morphological and Physiological Plant Traits

Green leaf area was evaluated using the LICOR LI-3100C leaf area meter (LI-COR, Inc., Lincoln, NE, USA). To assess root morphological traits, roots were extracted from the substrates and scanned using the WinRHIZO root-scanning equipment (Epson Expression 1100 XL, Epson America, Inc., Long Beach, CA, USA) and software (Regent Instruments, Inc. Quebec, QC, Canada).

### 4.5. Metabolite Extraction

In this study, we applied a modified method from Lundberg et al. [70] to extract metabolites. Briefly, samples were extracted from soil, clay, and sand on 21-day old sorghum plants by cutting the plant at the substrate line (if plant was present), removing the roots with rhizosphere soil attached, and placing roots into 10 mL of 70% methanol or high-performance liquid chromatography (HPLC) grade water contained within a 50-mL conical tube. The tube was shaken for ten seconds by hand and the roots were extracted and placed into a one-gallon bag with water for storage for root morphological analysis. The remaining bulk substrate from the plant treatment was then placed into a sanitized food processor and mixed for ten seconds on pulse. A five-gram subsample of the substrate was taken and placed into the respective 50-mL conical tube, that previously contained roots. The same process to collect a five-gram subsample of substrate was completed for bulk substrates from no-plant controls. Tubes were placed on a shaker on the tube’s side for two hours at 24 °C and centrifuged at 23 °C, 4750 ×
*g* for seven min. A two-mL sample of the liquid portion was placed into a microcentrifuge tube and the extract was evaporated using Thermo Savant^TM^ AES 2010 Speedvac^®^ system (Thermo Fisher Scientific, Waltham, MA, USA). Afterwards, the extract was resuspended by adding 100 µL of 70% methanol and briefly vortexed. The samples were divided for GC- and UPLC-MS analyses, with 50 µL transferred into respective microcentrifuge tubes for GC-MS, and the other 50 µL transferred into glass inserts in autosampler vials for UPLC-MS.

### 4.6. Metabolite Detection by Gas Chromatography—Mass Spectrometry

To prepare samples for GC-MS analysis, 50 μL of extract was dried using a speedvac, resuspended in 50 μL of pyridine containing 50 mg/mL of methoxyamine hydrochloride, incubated at 60 °C for 45 min, sonicated for 10 min, and incubated for an additional 45 min at 60 °C. Next, 25 μL of N-methyl-N-trimethylsilyltrifluoroacetamide with 1% trimethylchlorosilane (MSTFA + 1% TMCS, Thermo Scientific, Waltham, MA, USA) was added and samples were incubated at 60 °C for 30 min, centrifuged at 3000× *g* for 5 min, cooled to room temperature, and 80 μL of the supernatant was transferred to a 150 μL glass insert in a GC-MS autosampler vial. Metabolites were detected using a Trace GC Ultra coupled to a Thermo ISQ mass spectrometer (Thermo Scientific). Samples were injected in a 1:10 split ratio twice in discrete randomized blocks. Separation occurred using a 30 m TG-5MS column (Thermo Scientific, 0.25 mm i.d., 0.25 μm film thickness) with a 1.2 mL/min helium gas flow rate, and the program consisted of 80 °C for 30 seconds, a ramp of 15 °C per minute to 330 °C, and an 8 min hold. Masses between 50–650 m/z were scanned at 5 scans/sec after electron impact ionization.

### 4.7. Metabolite Detection by Ultra Performance Liquid Chromatography—Mass Spectrometry

For UPLC-MS analysis, 50 μL of extract was dried under nitrogen and resuspended in 100 μL of methanol. Then, 5 μL of extract was injected twice (*n* = 2 replicates) onto a Waters Acquity UPLC system in discrete, randomized blocks, and separated using a Waters Acquity UPLC HSS T3 column (1.8 µM, 1.0 × 100 mm), using a gradient from solvent A (water, 0.1% formic acid) to solvent B (Acetonitrile, 0.1% formic acid). Injections were made in 100% A, held at 100% A for 1 min, ramped to 98% B over 12 min, held at 98% B for 3 min, and then returned to starting conditions over 0.05 min and allowed to re-equilibrate for 3.95 min, with a 200 µL/min constant flow rate. The column and samples were held at 50 °C and 5 °C, respectively. The column eluent was infused into a Waters Xevo G2 Q-TOF-MS with an electrospray source in positive mode, scanning 50–1200 m/z at 0.2 sec per scan, alternating between MS (6 V collision energy) and MSE mode (15–30 V ramp). Calibration was performed using sodium formate with 1 ppm mass accuracy. The capillary voltage was held at 2200 V, source temperature at 150 °C, and nitrogen desolvation temperature at 350 °C with a flow rate of 800 L/hr.

### 4.8. Metabolomics Data Analysis

For each sample, raw data files were converted to .cdf format, and matrix of molecular features as defined by retention time and mass (m/z) was generated using XCMS software in R [71] for feature detection and alignment. Raw peak areas were normalized to total ion signal in R, outlier injections were detected based on total signal and PC1 of principle component analysis of mass binned XCMS peak areas and the mean area of the chromatographic peak was calculated among replicate injections (*n* = 2). Outliers were detected using Benjamini Hochberg corrected *p*-value returned by the R pnorm function. Molecular features were clustered using RAMClustR [72], which groups molecular features into spectra based on coelution and covariance across the full dataset, whereby spectra are used to determine the identity of observed compounds in the experiment (i.e., spectral clusters approximate individual compounds). The peak areas for each feature in a spectrum were condensed via the weighted mean of all features in a spectrum into a single value for each compound. Metabolites were annotated using RAMSearch software [73] and by searching against in-house and external metabolite databases including NIST v12, Massbank, Golm, and Metlin. A metabolite was annotated and assigned a confidence level of 1 if its spectral pattern and retention time matched that of an authentic standard analyzed in-house. We additionally compared the spectral pattern to that of an external database for further validation. A metabolite annotation was assigned a confidence level of 2 if the spectral pattern matched that of a public or theoretical spectral library. A chemical class annotation that resulted from a partial spectral match was assigned a confidence level of 3. Annotated compounds were grouped into the following chemical classes: carbohydrates, amino acids, organic acids, vitamins, and others [3,22], and reported with annotation confidence levels as previously described [74].

### 4.9. Statistical Analysis

Morphological traits were statistically analyzed by using an Analysis of Variance (ANOVA) for genotype, treatment, and their interaction using JMP Pro 11 (SAS Institute, Cary, NC, USA), followed by the Student’s *t*-test. Data were box-cox transformed prior to analysis in order to improve normality. Statistics assessing microbial presence were completed in JMP Pro 11 using ANOVA and a student’s *t*-test was computed to determine statistical significance among genotypes and substrates. Data were log transformed prior to analysis.

For metabolite statistical analysis, GC- and UPLC-MS data were combined and a principle components analysis (PCA) was performed using SIMCA v14.0 (Umetrics, Umea, Sweden) with unit variance (UV) scaling. Within each substrate, ANOVAs were performed by using the aov function in R (R Development Core Team, 2012). A false discovery rate (FDR) adjustment was used on the *p*-values using p.adjust function [75]. Log_2_ fold changes (FC) were calculated for each genotype by: log_2_ (plant treatment mean trait value/no-plant control mean trait value). Rhizosphere-enhanced metabolites (REMs) were those that were significant (*p* < 0.05) after applying the FDR adjustment and had a log_2_ FC of greater than one. Rhizosphere-abated metabolites (RAMs) were those that were significant after applying the FDR adjustment and had a log_2_ FC of less than negative one.

## 5. Conclusions

This study demonstrated an effective method to determine and quantify rhizosphere-associated metabolites involved in belowground plant–environment interactions using non-targeted metabolomics profiling. The intent of this study was to determine metabolites that are enriched or abated in the rhizosphere by the presence of the plant in substrates that represent more realistic field conditions and challenges. Future studies are required to explore the utility of this method in examining the functional roles of rhizosphere-associated metabolites in response to varying environmental conditions (abiotic and biotic stress) and within field soils. Overall, exploring root exudation in the context of the soil ecosystem will allow for a more accurate representation of the belowground plant–environment interaction and therefore may serve as a useful tool in designing more sustainable cropping systems.

## Figures and Tables

**Figure 1 ijms-20-00431-f001:**
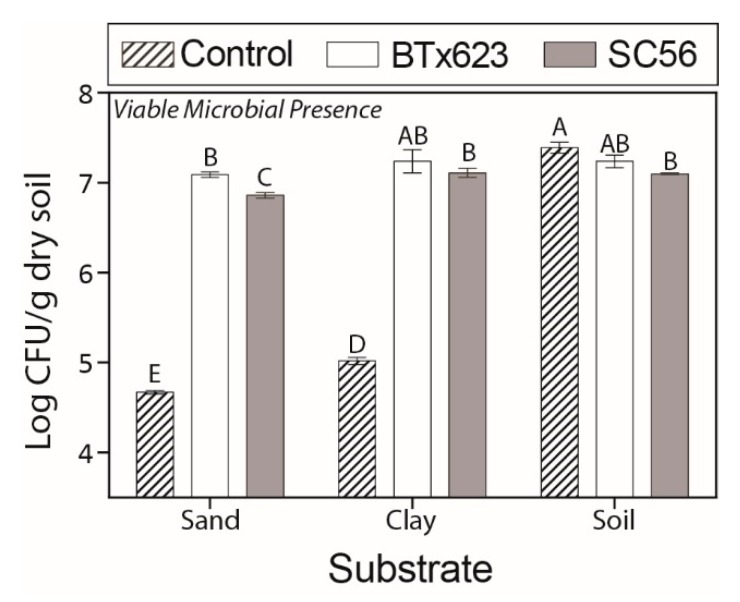
Viable microbial presence. Least square means and standard error of means (vertical bars) for the detected, culturable microorganisms for each treatment within each substrate. Uppercase letters indicate statistical significance (Student’s *t*) between means assessed among all substrates and genotypes.

**Figure 2 ijms-20-00431-f002:**
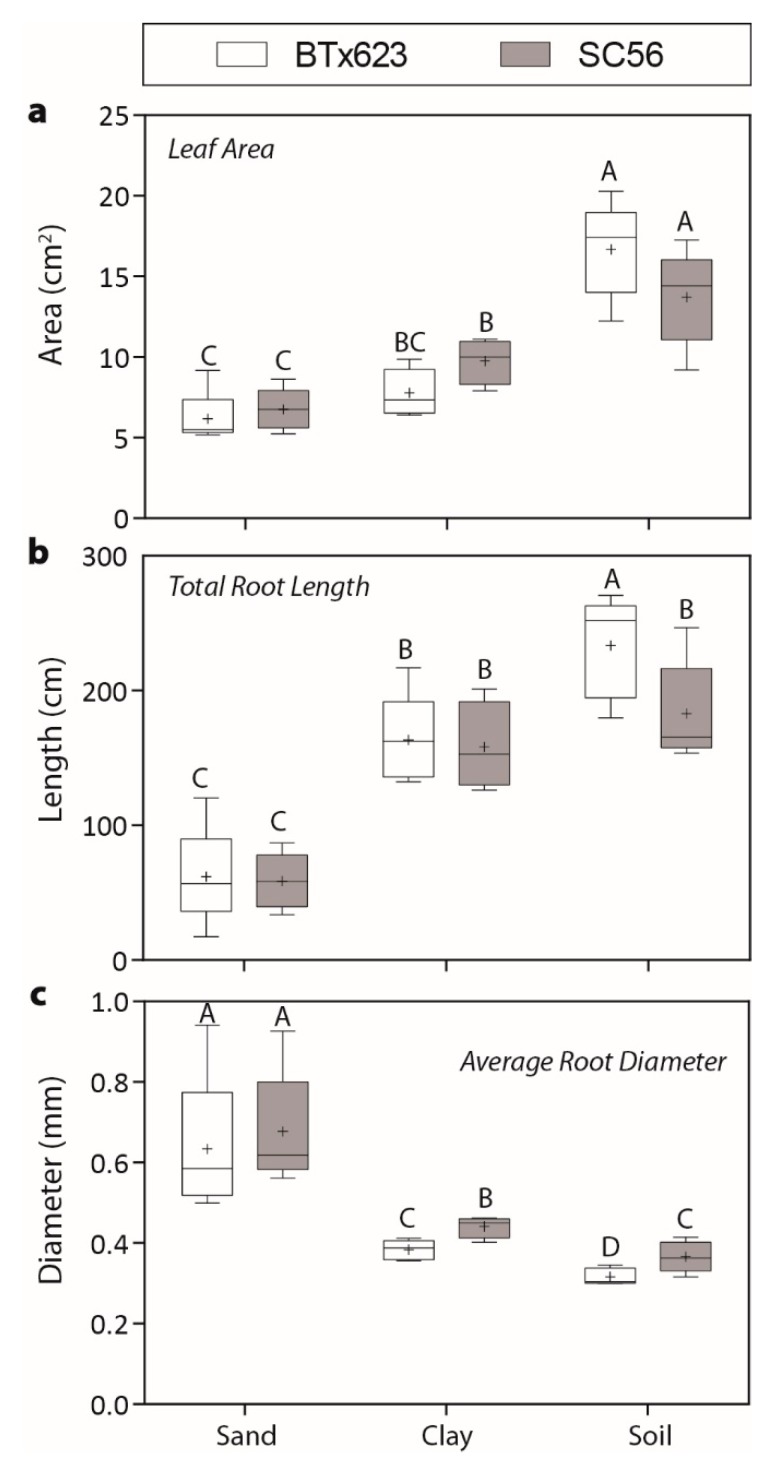
Morphological trait variation. Boxplots for (**a)** leaf area, (**b)** total root length, and (**c)** average root diameter. Boxplots represent median (line inside each box) and the bottom and top of boxes represent the lower and upper quartiles, respectively. The mean is indicated with a (+) and the bottom and top of each whisker represent the minimum and maximum of each observed trait, respectively. Uppercase letters indicate statistical significance (Student’s *t*) between means assessed within each trait measured.

**Figure 3 ijms-20-00431-f003:**
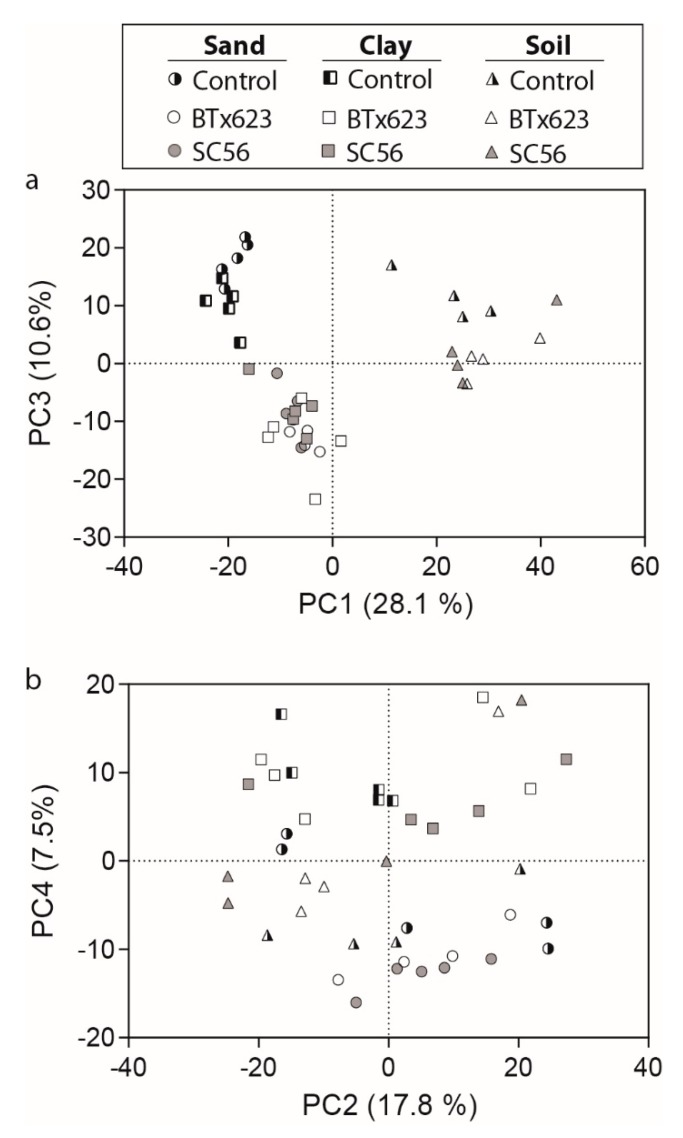
Principal component analysis (PCA) scores plot for principle components (**a**) 1 and 3 and (**b**) 2 and 4. Data from GC- and UPLC-MS analyses were combined, and the analysis is based on 1304 metabolites. No-plant controls are represented by half-shaded symbols, genotype BTx623 by open symbols, and genotype SC56 by closed symbols. Circles represent the sand substrate, squares the clay substrate, and triangles the soil substrate.

**Figure 4 ijms-20-00431-f004:**
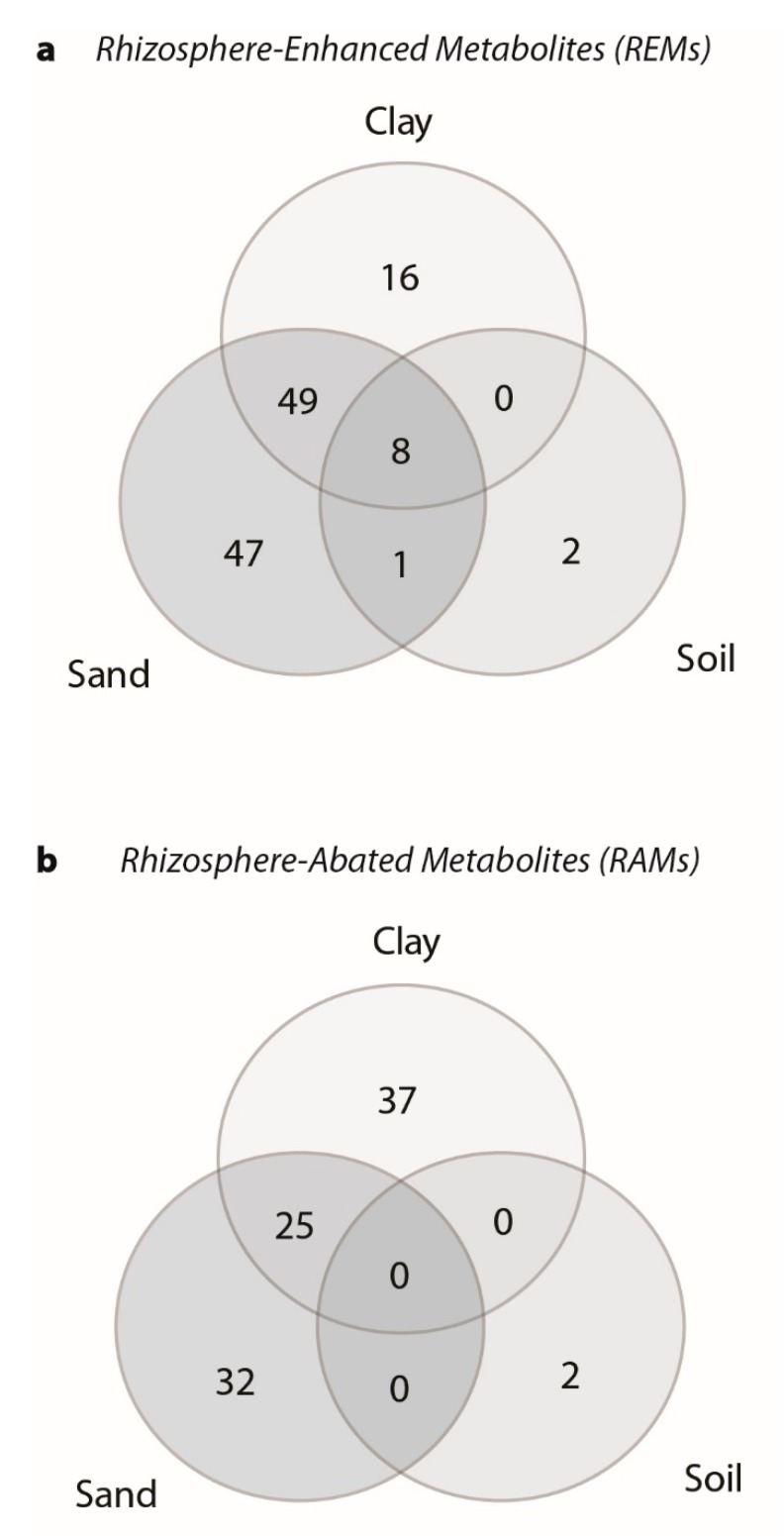
Venn diagram for the number of significant metabolites that were either (**a**) rhizosphere-enhanced metabolites (log_2_ FC > 1) or (**b**) rhizosphere-abated metabolites (log_2_ FC <−1). Shading indicates different substrates and the numbers in the overlapping regions represent the number of significant metabolites that are in common.

**Figure 5 ijms-20-00431-f005:**
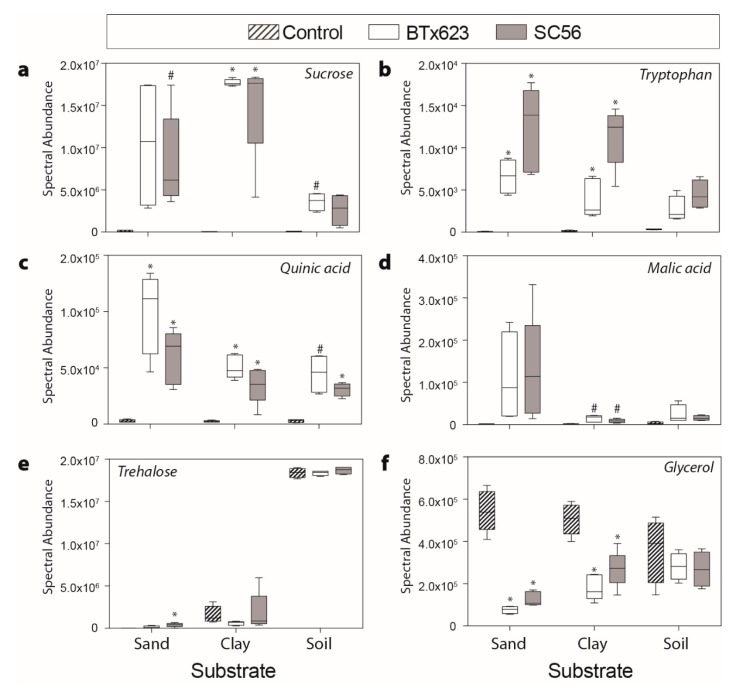
Boxplots of selected metabolites (**a**) sucrose, (**b**) tryptophan, (**c**) quinic acid, (**d**) malic acid, (**e**) trehalose, and (**f**) glycerol within clay, sand, and soil. Asterisk (*) indicates a significant difference between the genotype and control (*p* < 0.05) after false discovery rate adjustment; pound sign (#) indicates *p* < 0.10.

**Table 1 ijms-20-00431-t001:** Number of metabolites of interest detected within each substrate. Metabolites of interest were determined after adjusting p-values for false discovery rate and using *p* < 0.05 and a log_2_ fold change of >1 (REMs) or <−1 (RAMs).

Substrate	Total Metabolites of Interest	GC-MS	UPLC-MS	REMs	RAMs
Sand	162	119	43	105	57
Clay	135	113	22	73	62
Soil	13	9	4	11	2

**Table 2 ijms-20-00431-t002:** Annotated metabolites. List of annotated metabolites grouped by amino acids, carbohydrates, organic acids, vitamins, and others along with the platform detected, GC- or UPLC-MS and annotation confidence in parentheses. Metabolites that were annotated at a chemical class level are numbered if there are multiples (i.e., disaccharide 01, disaccharide 02). Associated log_2_ fold changes and false discovery rate (FDR) adjusted *p*-values for each genotype within each substrate are displayed. Bolded *p*-values are less than 0.1000.

		Sand				Clay				Soil			
Metabolite	Platform Detected	BTx623		SC56		BTx623		SC56		BTx623		SC56	
Amino Acids		*log_2_ FC*	*p-value*	*log_2_ FC*	*p-value*	*log_2_ FC*	*p-value*	*log_2_ FC*	*p-value*	*log_2_ FC*	*p-value*	*log_2_ FC*	*p-value*
alanine (2)	GC-MS	4.28	**0.0002**	4.24	**0.0820**	3.21	**0.0137**	2.71	**0.0150**	1.29	**0.0895**	1.05	0.3161
B-alanine (2)	GC-MS	3.67	**0.0012**	3.97	**0.0379**	1.37	0.4369	1.96	**0.0851**	0.78	0.1206	0.69	0.6791
aminobutyric acid (2)	GC-MS	2.50	**0.0026**	2.51	**0.0724**	0.87	0.3710	1.25	0.1438	0.80	0.2942	0.85	0.6579
glycine (2)	GC-MS	2.04	**0.0100**	2.02	**0.0617**	0.54	0.7521	1.08	**0.0631**	0.49	0.3633	0.52	0.7180
leucine (2)	UPLC-MS	5.25	**0.0001**	5.59	**0.0215**	4.26	**0.0258**	3.90	**0.0251**	0.87	0.3550	0.51	0.7694
phenylalanine (2)	UPLC-MS	7.48	**0.0256**	7.34	**0.0205**	4.56	**0.0134**	4.73	**0.0042**	1.43	0.2979	1.46	0.2670
pyroglutamate (2)	GC-MS	3.45	0.2487	2.57	0.4107	3.36	0.4140	1.43	0.2219	1.01	0.5871	2.09	0.4284
serine (2)	GC-MS	4.02	**0.0002**	3.85	**0.0219**	3.59	**0.0020**	2.90	**0.0261**	1.99	0.1103	1.76	0.2145
threonine (2)	GC-MS	4.42	**0.0010**	4.90	**0.0995**	2.45	0.3479	3.69	**0.0851**	0.71	0.2389	0.55	0.7657
tryptamine (2)	UPLC-MS	4.97	0.1013	5.22	**0.0213**	5.68	**0.0121**	5.70	**0.0113**	1.25	**0.0797**	2.36	0.1350
tryptophan (2)	UPLC-MS	6.88	**0.0033**	7.76	**0.0122**	4.73	**0.0469**	6.27	**0.0113**	2.96	0.2371	3.70	0.1547
tyrosine (2)	UPLC-MS	6.12	**0.0030**	5.47	**0.0280**	4.54	**0.0469**	4.15	**0.0127**	2.76	**0.0645**	1.92	0.3013
valine (2)	GC-MS	4.33	**0.0007**	4.40	**0.0811**	2.76	0.3001	3.71	**0.0241**	1.33	**0.0645**	0.90	0.7087
choline + glutamic acid (3)	UPLC-MS	5.20	**0.0065**	5.04	**0.0018**	2.40	**0.0145**	2.19	**0.0259**	1.35	**0.0645**	1.06	0.1726
C_5_H_11_NO_2_ (valine) (3)	UPLC-MS	5.14	**0.0008**	2.45	**0.0379**	0.63	0.3200	0.53	0.2153	−0.88	0.5546	−1.59	0.4187
**Carbohydrates**													
fructose (2)	GC-MS	7.72	**0.0008**	7.14	**0.0006**	7.29	**0.0034**	6.80	**0.0241**	2.06	0.1461	2.18	0.2888
glucose (2)	GC-MS	8.09	**0.0042**	7.88	**0.0067**	6.65	**0.0025**	6.11	**0.0247**	0.24	0.2436	0.29	0.7694
glycerol (2)	GC-MS	−2.84	**0.0010**	−2.10	**0.0015**	−1.48	**0.0041**	−0.90	**0.0261**	−0.49	0.7143	−0.38	0.7425
myo-inositol (2)	GC-MS	4.60	**0.0006**	4.96	**0.0004**	4.39	**0.0120**	4.03	**0.0113**	0.31	0.2587	0.30	0.4533
sucrose (2)	GC-MS	6.53	0.1135	6.20	**0.0666**	8.56	**<0.0001**	8.32	**0.0200**	5.21	**0.0635**	4.93	0.3333
trehalose (2)	GC-MS	3.95	0.2333	5.51	**0.0489**	−1.46	0.2067	0.24	0.9132	−0.33	0.9717	0.02	0.7694
disaccharide 01 (3)	UPLC-MS	6.01	**<0.0001**	5.54	**0.0101**	3.04	**0.0343**	2.69	**0.0289**	0.49	0.5546	0.40	0.5050
disaccharide 02 (3)	UPLC-MS	6.90	**0.0005**	5.09	**0.0080**	1.23	**0.0293**	1.59	**0.0560**	−0.49	0.6437	−0.16	0.8950
hexose sugar acid (3)	GC-MS	1.10	**0.0330**	0.70	0.1017	4.13	**0.0093**	3.66	**0.0627**	1.15	0.1156	1.24	0.4533
hexose + glutamine (3)	UPLC-MS	5.93	**<0.0001**	4.97	**0.0010**	3.48	**0.0137**	3.17	**0.0259**	0.94	0.1785	0.33	0.5323
hexose 01 (3)	GC-MS	7.54	**0.0013**	7.56	**0.0026**	6.24	**0.0190**	5.60	**0.0498**	0.05	0.2892	0.12	0.7679
hexose 02 (3)	GC-MS	4.39	**0.0012**	4.04	**0.0051**	3.93	**0.0254**	3.70	**0.0368**	−0.08	0.7143	0.04	0.9964
inositol-like (3)	GC-MS	3.31	**0.0508**	2.84	**0.0121**	1.91	**0.0237**	1.41	**0.0188**	−0.07	0.6744	0.10	0.8236
pentose (3)	GC-MS	3.67	**0.0004**	3.54	**0.0158**	3.74	**0.0025**	3.60	**0.0244**	1.05	0.1785	0.99	0.3661
sugar alcohol 01 (3)	GC-MS	5.16	0.1614	5.68	**0.0486**	3.25	0.5230	3.94	0.2763	−0.13	0.8860	0.65	0.8556
sugar alcohol 02 (3)	GC-MS	5.94	0.2057	7.22	**0.0382**	0.19	0.9370	1.64	0.3052	−.31	0.1206	−1.75	0.0286
sugar alcohol 03 (3)	GC-MS	0.58	0.2693	0.37	0.2932	0.89	0.2100	0.64	**0.0955**	0.69	0.1557	2.01	0.2259
trisaccharide (3)	GC-MS	−0.21	0.7571	0.07	0.9323	−0.67	0.3383	−0.58	0.3475	0.49	0.6368	2.01	0.4472
**Organic Acids**													
aconitic acid (2)	GC-MS	4.51	0.5198	1.70	**0.0802**	0.44	0.8463	0.54	0.7111	2.18	0.6733	1.74	0.4837
glyceric acid (2)	GC-MS	1.59	**0.0198**	0.99	0.1468	2.29	**0.0818**	1.50	**0.0749**	0.55	0.3271	0.77	0.4444
malic acid (2)	GC-MS	6.02	0.2159	6.25	0.1816	3.17	**0.0559**	2.43	**0.0599**	2.39	0.4720	2.07	0.2145
quinic acid (2)	GC-MS	5.11	**0.0085**	4.36	**0.0136**	4.40	**0.0015**	3.85	**0.0276**	3.53	**0.0927**	3.25	**0.0320**
threonic acid (2)	GC-MS	5.42	**0.0053**	5.68	**0.0234**	5.88	**0.0015**	5.82	**0.0379**	3.01	0.1206	2.68	0.1350
**Vitamins**													
pantothenic acid (2)	UPLC-MS	5.44	**0.0269**	4.72	**0.0176**	5.31	**0.0249**	4.49	**0.0181**	4.64	0.1206	3.78	0.1753
**Other**													
dhurrin (2)	UPLC-MS	8.27	**0.0235**	7.59	**0.0214**	7.33	**0.0015**	6.50	**<0.0001**	5.55	**0.0348**	5.11	**0.0286**
prolyl-histidine-like (3)	UPLC-MS	2.41	0.2219	2.07	**0.0529**	8.23	**0.0134**	8.65	**0.0045**	0.84	**0.0927**	0.97	0.1753
tyrosyl-histidine-like (3)	UPLC-MS	7.98	**0.0079**	6.81	**0.0200**	6.77	**0.0172**	5.01	**0.0149**	5.49	**0.0645**	3.63	0.2259

It should be noted that the annotated metabolites represent a portion of the varying metabolites within each substrate, and not all of the annotated metabolites were statistically significant in every substrate (Table 2). There were many other varying metabolites that were unable to be annotated by spectral matching to the major plant metabolite databases. These unannotated metabolites displayed consistent trends across the substrates. We present a subset of annotated metabolites that were rhizosphere-enhanced metabolites to include two sugars (sucrose, trehalose), an amino acid (tryptophan), and organic acids (quinic acid, malic acid) (Figure 5). In addition, we provide an example of a metabolite that was a rhizosphere-abated metabolite (glycerol).

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
