# Peer review of "Non-Targeted Metabolomics Reveals Sorghum Rhizosphere-Associated Exudates are Influenced by the Belowground Interaction of Substrate and Sorghum Genotype"

_ijms, 2019, doi:10.3390/ijms20020431_

Round 1
Reviewer 1 Report
In this study authors utilized non-targeted metabolomics to investigate how differing substrate conditions and genotypic background drive variation in metabolites exudated by sorghum.
Authors used a different approach, by using more realistic substrates instead of artificial systems (e.g. hydroponic systems or sterile media). Furthermore, using two genotypes of sorghum (BTx623 and SC56), various growth substrates (soil, clay and sand) and a non-targeted metabolomics analysis (GC- and UPLC-MS), authors identify 263 quantitative differences in metabolites among genotypes and substrates.
If we considered that the rhizosphere as the narrow region of soil that is directly influenced by root secretions and associated soil microorganisms, both types of components need to be quantify and qualify to clearly assess its contribution. On this work, authors used as experimental design a comparison between compounds extracted from roots on different substrates and the exudate present in the substrate without a plant (no-plant control). They argue that autoclave the substrate may change its characteristics (pH, cation capacity, etc.) but, either literature or results on this paper do not support this statement. However, which it seems to be relevant for agricultural purposes is to identify the effects of the rhizosphere in plant growth (increase resistance to stresses and/or biomass).
Authors used on this work, selected two sorghum varieties, BTx623 is a sequenced genotype that is pre-flowering drought 453 tolerant and SC56 is a pre-flowering drought susceptible genotype. Authors also realized morphological and physiological plant traits quantifications (root length, diameter and leaf area), which were not sufficiently contributing to the interpretation of the results.
This work contain important data but may be presented, analyzed and interpreted using improved experimental design to contribute to better understanding the importance of the rhizosphere in plant
Proposed Modifications
- Change the order of results presentation
2.1 Soil composition + Viable Microbial presence (2.4)
Combine both in a first result section as a clear description of the different substrates
Table 1 may be removed and place on Supp data.
2.2 Variation in plant morphology
2.3 Non-targeted Metabolomics
2.4 Correlation analysis
- I propose to include a section about on which authors compare genotypes on different substrates (without excluding metabolites present on no-plant samples). It may be interesting to compare the selected genotypes on different substrates and discuss about it implication on their phenotypes (tolerant BTx623 and susceptible against drought, SC56).
- Discussion maybe more precise and pertinent (reduce to half).
Explanation:
Line 122 Substrates were not autoclaved as the heat, steam, and pressure are expected to alter substrate characteristics.
Do you have a reference for this?
Line 123 Additionally, we evaluated two extraction buffers, 70% methanol and 100% HPLC grade water.
Line 144 The water extraction resulted in overall very low metabolite diversity and signal intensity and was determined to be insufficient for metabolomics analysis (data 145 not shown).
Water extraction was part of the optimization of the methodology used on this work. If any results are presented, I suggest removing these sentences from the manuscript (including in the Discussion section 3.5)
Line 147 The GC- and UPLC-MS analyses resulted in 34,718 and 2,929 molecular features [35] that were deconvoluted into an estimated 829 and 475 compounds, respectively [36].
Reference may be removed from here (part of material and methods)
Can authors indicate how many compounds were only detected on control samples?
Are those values correlated with the present of viable microbial on each type of soil?
Line 160 Individual metabolites that varied due to each plant treatment (BTx623 and SC56)
Change to plant genotype
Line 168 (BTx623 and SC56) and no-plant control for all detected metabolites are displayed as volcano plots (Fig. S1).
This type of analysis maybe done also to compare different genotype on the same substrate
Author Response
Response to Reviewer 1
Change the order of results presentation
1. 2.1 Soil composition + Viable Microbial presence (2.4). Combine both in a first result section as a clear description of the different substrates.
2. Table 1 may be removed and place on Supp data.
We changed the order of results. The soil composition and viable microbial presence were combined and placed into the first section (2.1). Additionally, we have moved the substrate composition table into supplemental data.
I propose to include a section about on which authors compare genotypes on different substrates (without excluding metabolites present on no-plant samples). It may be interesting to compare the selected genotypes on different substrates and discuss about it implication on their phenotypes (tolerant BTx623 and susceptible against drought, SC56).
We agree that comparing genotypes directly and discussing the implications on the phenotypes would be beneficial. However, very few, metabolites were found to be significant when solely comparing the genotypes, likely due to large plant-to-plant variation (biological variation) within each plant genotype treatment (see Figure 5). This lack of significance is addressed in the discussion by suggesting using large numbers of replicates and standardizing samples prior to GC- and UPLC-MS using total root length to account for differences in size.
Discussion maybe more precise and pertinent (reduce to half).
We have significantly reduced and focused the discussion section.
Line 122. Substrates were not autoclaved as the heat, steam, and pressure are expected to alter substrate characteristics. Do you have a reference for this?
Added references in this section that were also included in the introduction (Line 65-68 and Line 118)
Line 123. Additionally, we evaluated two extraction buffers, 70% methanol and 100% HPLC grade water.
This has been removed.
Line 144 The water extraction resulted in overall very low metabolite diversity and signal intensity and was determined to be insufficient for metabolomics analysis (data 145 not shown).
Water extraction was part of the optimization of the methodology used on this work. If any results are not presented, I suggest removing these sentences from the manuscript (including in the Discussion section 3.5)
This has been removed.
Line 147 The GC- and UPLC-MS analyses resulted in 34,718 and 2,929 molecular features [35] that were deconvoluted into an estimated 829 and 475 compounds, respectively [36].
Reference may be removed from here (part of material and methods)
This has been removed.
Can authors indicate how many compounds were only detected on control samples? Are those values correlated with the present of viable microbial on each type of soil?
We did initially compare control samples, but felt the information was outside the scope to include in this manuscript. We acknowledge that correlations are a good future direction, but we did not feel comfortable conducting correlations as our sample size was small and therefore confidence in these data is low. Additionally, many factors beyond metabolites such as other soil characteristics influence microbial abundances which could confound the results.
Line 160
Individual metabolites that varied due to each plant treatment (BTx623 and SC56)
Change to plant genotype
Changed to plant genotype.
Line 168 (now …
(BTx623 and SC56) and no-plant control for all detected metabolites are displayed as volcano plots (Fig. S1). This type of analysis maybe done also to compare different genotype on the same substrate.
We agree that volcano plots comparing the different genotypes on the same substrate would make sense if there were significant p-values between the genotypes. However, very few metabolites were found to be significant when solely comparing the genotypes, and volcano plots did not aid in the determination of metabolites of interest between genotypes for this study.
Reviewer 2 Report
This manuscript describes a well designed and executed untargeted metabolomic comparison of two sorghum genotypes in three different growth matrices. It is well crafted, and does not need any major editing or changes. Some minor comments/questions that could be addressed in a final draft:
I feel that figure one would be more informative as a boxplot, and perhaps explain why box-cox transformation was used in the statistical analyses of the plant morphology data.
It was particularly nice that the p-values in the study were FDR adjusted.
In tables 2 and 3 it appears that there were no compounds in common between the GC and the LC methods. Is that true? It would be quite nice to have an orthogonal validation of some of the metabolites by seeing them in both methods.
In table 3 one of your column headings is missing "FC" (says just log2)
In figure 3's caption you state "colors indicate" when it is a greyscale figure. I do not think it needs to be color, just a correction of the figure. It is quite clear as is.
Line 243 "levelss" should be "levels"
The section on outlier identification (using TIC and PC1) could use a bit more information. What were the exclusion criteria for each test? How many samples were excluded? What was done in case of exclusion? Was the sample reinjected, the plant reextracted?
Author Response
Response to Reviewer 2
I feel that figure one would be more informative as a boxplot, and perhaps explain why box-cox transformation was used in the statistical analyses of the plant morphology data.
We have changed Figure 1 to a boxplot.
It was particularly nice that the p-values in the study were FDR adjusted.
In tables 2 and 3 it appears that there were no compounds in common between the GC and the LC methods. Is that true? It would be quite nice to have an orthogonal validation of some of the metabolites by seeing them in both methods.
LC-MS and GC-MS are orthogonal methods given the extractions we used and the type of samples we are analyzing. The compounds that overlap would be in the unretained fraction of LC-MS data, and therefore would not provide good quantitative information due to ion suppression for these types of metabolites. For the nonpolar and semi-polar compounds detected by GC-MS, these were likely below the limit of detection on our LC equipment given they were not detected in the data set.
In table 3 one of your column headings is missing "FC" (says just log2)
We have made this correction.
In figure 3's caption you state "colors indicate" when it is a greyscale figure. I do not think it needs to be color, just a correction of the figure. It is quite clear as is.
We have changed the word “color” to “shading” for clarity.
Line 243 "levelss" should be "levels"
This has been corrected.
The section on outlier identification (using TIC and PC1) could use a bit more information. What were the exclusion criteria for each test? How many samples were excluded? What was done in case of exclusion? Was the sample reinjected, the plant reextracted?
We acknowledge that this section may be unclear and therefore expanded on text and included that outlier testing was performed on the total XCMS detected signal and the first principle component of mass binned XCMS peak areas. Each extraction was injected multiple times, and outliers injections were detected using Benjamini Hochberg corrected p-value returned by the R pnorm function.